# Utilization of Whey for Eco-Friendly Bio-Preservation of Mexican-Style Fresh Cheeses: Antimicrobial Activity of *Lactobacillus casei* 21/1 Cell-Free Supernatants (CFS)

**DOI:** 10.3390/ijerph21050560

**Published:** 2024-04-28

**Authors:** Victor E. Vera-Santander, Ricardo H. Hernández-Figueroa, Daniela Arrioja-Bretón, María T. Jiménez-Munguía, Emma Mani-López, Aurelio López-Malo

**Affiliations:** Department Ingeniería Química, Alimentos y Ambiental, Universidad de las Américas Puebla, Ex-Hacienda Santa Catarina Mártir S/N, San Andrés Cholula 72810, Puebla, Mexico; victor.verasr@udlap.mx (V.E.V.-S.); ricardoh.hernandezf@udlap.mx (R.H.H.-F.); emma.mani@udlap.mx (E.M.-L.)

**Keywords:** cell-free supernatants, dairy industry, food safety, natural antimicrobials, lactic-acid bacteria, sustainability

## Abstract

Using whey, a by-product of the cheese-making process, is important for maximizing resource efficiency and promoting sustainable practices in the food industry. Reusing whey can help minimize environmental impact and produce bio-preservatives for foods with high bacterial loads, such as Mexican-style fresh cheeses. This research aims to evaluate the antimicrobial and physicochemical effect of CFS from *Lactobacillus casei* 21/1 produced in a conventional culture medium (MRS broth) and another medium using whey (WB medium) when applied in Mexican-style fresh cheese inoculated with several indicator bacteria (*Escherichia coli*, *Salmonella enterica* serovar Typhimurium, *Staphylococcus aureus*, and *Listeria monocytogenes*). The CFSs (MRS or WB) were characterized for organic acids concentration, pH, and titratable acidity. By surface spreading, CFSs were tested on indicator bacteria inoculated in fresh cheese. Microbial counts were performed on inoculated cheeses during and after seven days of storage at 4 ± 1.0 °C. Moreover, pH and color were determined in cheeses with CFS treatment. Lactic and acetic acid were identified as the primary antimicrobial metabolites produced by the *Lb. casei* 21/1 fermentation in the food application. A longer storage time (7 days) led to significant reductions (*p* < 0.05) in the microbial population of the indicator bacteria inoculated in the cheese when it was treated with the CFSs (MRS or WB). *S. enterica* serovar Typhimurium was the most sensitive bacteria, decreasing 1.60 ± 0.04 log_10_ CFU/g with MRS-CFS, whereas WB-CFS reduced the microbial population of *L. monocytogenes* to 1.67 log_10_ CFU/g. *E. coli* and *S. aureus* were the most resistant at the end of storage. The cheese’s pH with CFSs (MRS or WB) showed a significant reduction (*p* < 0.05) after CFS treatment, while the application of WB-CFS did not show greater differences in color (ΔE) compared with MRS-CFS. This study highlights the potential of CFS from *Lb. casei* 21/1 in the WB medium as an ecological bio-preservative for Mexican-style fresh cheese, aligning with the objectives of sustainable food production and guaranteeing food safety.

## 1. Introduction

Most traditional Mexican fresh cheeses are artisanal products made from whole or low-fat cow’s milk, frequently without the milk’s thermal treatment, through casein coagulation with rennet (or other proteinase substitutes) [1]. Their processing is not commonly standardized; producers handle low production volumes and often utilize unpasteurized milk [2], which represents a health risk [3]. In the production of fresh Mexican-style cheese, no bacteria are inoculated because it is not a fermented product; due to its short shelf life, it is usually consumed soon after it is made [2]. It is common for this variety of cheese to be a vehicle for pathogenic microorganisms, which can be attributed to unpasteurized milk or handling techniques during its production [4]. Unfortunately, there have been multiple cases of illness linked to the consumption of this cheese, as it may contain harmful microorganisms such as *Salmonella* [5,6,7], *Listeria monocytogenes* [8,9,10], *Escherichia coli* [11,12], and *Staphylococcus aureus* [6,13,14].

Bio-preservation involves using microorganisms or their metabolites to increase the shelf life and safety of food. Among of the most promising natural biological antagonists are lactic acid bacteria (LAB), which have several potential applications. Numerous studies have reported that LAB strains produce antimicrobial substances (organic acids, short-chain fatty acids, hydrogen peroxide, and proteinaceous compounds) [15,16]. LAB are commonly used as starter cultures in the food industry, particularly in cultured dairy products [17]. Nonetheless, non-starter LAB have also been assessed for their ability to prolong the shelf life of various foods, including bread, dairy products, fresh fruits, vegetables, and animal feed [18,19,20]. Several studies have investigated the potential use of non-traditional starter LAB to increase cheese safety and shelf life. Dal Bello et al. [21] successfully controlled the growth of *L. monocytogenes* in cottage cheese using LAB. Settanni et al. [22] were able to extend the safety and shelf life of Tosela cheese by controlling the growth of *L. monocytogenes*, *Salmonella* spp., and coliforms using *Lactobacillus paracasei* NdP78 and *Streptococcus macedonicus* NdP1.

LAB are generally recognized as safe (GRAS) with qualified presumption of safety (QPS) [23]; many of them have been studied as bio-preservatives due to the secretion of antimicrobial compounds, either organic acids or bacteriocins [20]. Ensuring the safety of cheeses while maintaining their organoleptic characteristics is a challenge for the food industry. Therefore, the use of LAB cell-free supernatants (CFSs) can be an alternative since they have interesting antimicrobial properties [19], especially in the bio-preservation of raw foods such as beef [24], chicken [25,26,27], shrimp [28], and vegetables [29], among many others [20].

To produce these CFSs and enable their possible use in the food industry, it is essential to have a culture medium for the optimal growth of microorganisms and the production of antimicrobial metabolites. It is important to consider the economic viability and safety for consumption of the medium used to produce CFS. In most studies [19] addressing the antimicrobial activity of CFS from LAB, the culture medium used is de Man, Rogosa, and Sharpe (MRS) broth, which is excellent for the growth of *Lactobacillus*. However, due to its high cost, MRS broth is unsuitable for industrial use. Zandona et al. [30] have mentioned that whey is a sustainable food ingredient since it has been employed in the elaboration of food products such as infant formula, meat products, beverages, soups, sauces, toppings, creamers, nut coatings, pressed nuts, cheese-based sauces, potato chips, savory flavors, savory puff pastries, and special bakery products such as pizza, biscuits, macaroni, soufflés, and cakes. Hence, whey may be a potential ingredient for food-grade culture medium since its nutrient profile includes sugar (lactose 46–52 g L^−1^), proteins (6.5–6.6 g L^−1^), and minerals (5.0–5.2 g L^−1^) [31]. Recently, fermented whey by LAB has been investigated as an antimicrobial. For instance, Yousefi et al. [32] optimized the fermentation conditions of *L. plantarum* PTCC 1896 in whey to improve the antibacterial metabolites production, whereas Izzo et al. [33] analyzed the antifungal activity from fermented goat’s sweet whey utilizing four *L. plantarum* strains. However, they did not evaluate the antimicrobial activity in any food matrix, and thus, further research is necessary. Also, whey has been demonstrated to be suitable for LAB growth and fermented probiotic beverages [31,34]. Whey derived from the cheese industry is relatively cheap and readily available. This approach would help improve food safety and the environment since the cheese industry produces approximately 115 million tons, wasting around 47% [35]. Discharging untreated whey into the environment can cause several problems such as water contamination, dissolved oxygen depletion, and eutrophication [36]. Whey dissolved in water has a high level of biochemical oxygen demand (BOD) ranging from 40 to 60 g L^−1^ and chemical oxygen demand (COD) ranging from 50 to 80 g L^−1^ [30].

Therefore, this research aims to evaluate the antimicrobial and physicochemical effect of CFS from *Lactobacillus casei* 21/1 produced in a conventional culture medium (MRS broth) and another medium using whey (WB) when applied in Mexican-style fresh cheese inoculated with several indicator bacteria (*Escherichia coli*, *Salmonella enterica* serovar Typhimurium, *Staphylococcus aureus*, and *Listeria monocytogenes*).

## 2. Materials and Methods

Figure 1 shows the flow diagram of the experiments performed in the present study.

### 2.1. Bacterial Strains, Culture Mediums Preparation and Growth Conditions

*Lactobacillus casei* 21/1 as well as indicator strains (*Escherichia coli* ATCC 25922, *Salmonella enterica* serovar Typhimurium ATCC 14028, *Staphylococcus aureus* ATCC 29213, and *Listeria monocytogenes* Scott A) were obtained from the Food Microbiology Laboratory of the Universidad de las Americas Puebla (San Andres Cholula, Puebla, Mexico). *Lb. casei* 21/1 was reactivated and sub-cultivated in de Man, Rogosa, and Sharpe (MRS) broth (Difco™ BD, Sparks, MD, USA) at 35 °C. Indicator strains were reactivated and incubated in trypticase soy broth (TSB, Bioxon BD, Mexico City, Mexico) at 35 ± 1.0 °C for 24 h.

The whey-based (WB) medium was prepared using 10.0% *w*/*w* whey powder (10% protein, 1.5% fat, 75% carbohydrates, and 1.1% sodium chloride; Food Technologies Trading, Mexico City, Mexico), yeast extract 0.3% *w*/*w* (Difco™, BD, Sparks, MD, USA), magnesium sulfate 0.02% (MgSO_4_•7H_2_O) (Merck, Burlington, MA, USA), manganese sulfate 0.005% *w*/*w* (MnSO_4_•H_2_O) (Merck, Burlington, MA, USA), and water. In a previous study [37], MRS broth (Difco™ BD, Sparks, MD, USA) was used as a conventional culture medium. Fresh cultures of *Lb. casei* 21/1 were used to inoculate the culture media, adding the necessary amount to obtain an initial population of 10^6^ CFU mL^−1^. For the growth conditions in culture media, *Lb. casei* 21/1 was incubated at 35 ± 1.0 °C for 48 h.

### 2.2. Preparation of Cell-Free Supernatant (CFS)

After fermentation with *Lb. casei* 21/1, the MRS broth and WB medium were centrifugated at 7000× *g* for 15 min at 5 °C (Sorvall ST 8R, Thermo Fischer Scientific, Schwerte, Germany). Later, the supernatants were filtered through a 0.45 μm Millipore membrane filter (Advantec, MFS, Dublin, CA, USA). The CFSs from MRS or WB were concentrated 10-fold via vacuum evaporation on a Buchi R-210/215 rotary evaporator (Buchi, Flawil, Switzerland) at 70 ± 1.0 °C and 25 cm Hg following the methodology described by Arrioja-Bretón et al. [37]. Concentrated CFSs were refrigerated at 4 ± 1.0 °C until their use.

### 2.3. Characterization of CFSs

#### 2.3.1. pH and Titratable Acidity

The pH of MRS and WB CFSs was determined by immersion electrode using a pH meter (HI 2210 Hanna Instruments, Woonsocket, RI, USA). The percentage of titratable acidity (TA%) of CFSs was determined following method 22.061 from AOAC [38] and was expressed as a percentage of lactic acid (%*w*/*v*). The measurements were performed in triplicate.

#### 2.3.2. Organic Acid Determination in CFSs

The organic acids concentration from MRS and WB CFSs (lactic and acetic acid) was determined via high-performance liquid chromatography (HPLC) following the methodology reported by Hernández-Figueroa et al. [39]. The chromatograph employed was an Agilent 1260 (Agilent Technologies, Santa Clara, CA, USA) coupled with a diode array detector (DAD) at a wavelength of 210 nm. CFSs (MRS or WB) were injected (20 µL) using an Agilent G1329 autosampler (Agilent Technologies, Santa Clara, CA, USA). A C-18 column (250 × 4.6 mm) (Restek, Centre Country, PA, USA) was used with a mobile isocratic phase of 20 mM monobasic potassium phosphate buffer solution (adjusted to pH 2.4) at 0.6 mL min^−1^ at room temperature. For the quantification of lactic and acetic acid, the concentration of acids was linearly correlated with their respective peak areas, obtaining correlation coefficients R^2^ > 0.99 employing standard solutions.

### 2.4. Laboratory-Scale Cheese Production

For the food application of CFSs, Mexican-style fresh cheese was manufactured with commercial pasteurized brand Lala™ (Gomez Palacio, Durango, Mexico) whole milk (fat: 3.3%, protein: 3.1%, carbohydrates: 4.7%, 5 g of vitamin D/L, and 666 mg retinol equivalents/L) following the methodology of Parra-Ocampo et al. [40] with slight modifications. The milk was heated until reaching a temperature of 39 ± 1.0 °C, then 0.02% of 50% calcium chloride (Reactivos Química Meyer, Mexico City, Mexico) and 0.03% of commercial chymosin (Cuamex^®^, Chr. Hansen, Mexico City, Mexico) per liter of milk were added and stirred. Then, the milk was left to rest for 20 min until it curdled. The curd was cut into cubes of approximately 2 cm^3^ and left to rest for another 20 min. Cubes were subsequently placed on a mesh and squeezed to remove the whey. To the resulting paste, 1% salt was added to the total cheese mass. Finally, the cheese mass was molded in a cylindrical mold and pressed at 4 ± 1.0 °C for 24 h. Different batches of cheese (≈10 pieces each) were prepared for all the treatments and analyses that were carried out.

### 2.5. Physicochemical Analyses

#### 2.5.1. Proximate Analysis

For characterization of manufactured Mexican-style fresh cheese, AOAC [41] methods were taken as references to determine moisture (method 33.7.03), fat (method 933.05), ash (method 33.7.07), and protein (method 33.7.12 via the Kjeldahl method using a conversion factor of 6.38). These determinations were performed in triplicate.

#### 2.5.2. pH and Color

The cheese pH was measured according to NMX-F-317-S-1978 [42]; 10.0 ± 1.0 g of non-inoculated cheese (non-CFS) and CFS treated (MRS and WB) were mixed with 50 mL distilled water. An electrode connected to a digital pH meter (HI 2210 Hanna Instruments, Woonsocket, RI, USA) was employed. A Konica Minolta CR-400 colorimeter (Konica Minolta, Tokio, Japan) was used in reflectance mode and CIELAB scale to determine the color of the samples. Color differences (ΔE) were assessed using Equation (1) where $L_{0}^{*}\;$ and $L^{*}$ are the initial luminosity of the sample and at the time it was analyzed, $a_{0}^{*}$ and $a^{*}$ are the initial red–green contribution, and $b_{0}^{*}$ and $b^{*}$ are the initial blue–yellow contribution of the initial sample and the analyzed sample at each time point, respectively. The pH and color measurements were taken in triplicate at 0, 4, and 7 days of storage at 4 ± 1.0 °C.
(1)$$\mathrm{the}\;\Delta E=\sqrt{{\left(L_{0}^{*}-L^{*}\right)}^{2}+{\left(a_{0}^{*}-a^{*}\right)}^{2}+{\left(b_{0}^{*}-b^{*}\right)}^{2}\;}$$

### 2.6. Cheese Inoculation and CFS Treatment

The indicator bacteria (*E. coli*, *S. aureus*, *S. enterica* serovar Typhimurium, and *L. monocytogenes*) were cultivated according to the conditions described in Section 2.1. Adequate 10-fold dilutions were performed to obtain a cell suspension of 10^5^ CFU mL^−1^. Each side of a piece of Mexican-style fresh cheese manufactured in the laboratory (~10 g) was inoculated with 250 μL of the inoculum of each indicator microorganism and drained for 20 min. Subsequently, cheese pieces were put in individual plastic bags (Whirl-Pak1, Nasco, Fort Atkinson, WI, USA) with 1 mL of CFS from *Lb. casei* 21/1 (from MRS or WB) for surface spread. Then, the samples were stored at 4 ± 1.0 °C for 7 days. Inoculated indicator bacteria without CFSs cheese pieces were used as a negative control. These determinations were performed in triplicate.

### 2.7. Microbiological Analysis of Laboratory-Scale and Locally Marketed Cheeses

After the application of CFSs in cheese, microbial counts were performed for each indicator microorganism on days 0, 4, and 7; one piece (~10 g) of cheese inoculated with each bacterium was put in a sterile plastic bag (Whirl-Pak1, Nasco, Fort Atkinson, WI, USA), and homogenized for 2 min in a Stomacher 80 lab blender (Seward Ltd., West Sussex, UK) with 90 mL of sterile peptone water (1 g L^−1^). Adequate decimal dilutions were prepared into peptone water and plated in trypticase soy agar. Plates were incubated during 18–24 h at 37 ± 1.0 °C.

Mexican-style fresh cheeses are commonly consumed, and it is well known that they are associated with foodborne diseases [6]. For this reason, this study investigated the ability of the CFSs to help reduce the risk associated with these products. For comparison purposes, two cheeses were purchased in local markets (San Andres Cholula, Puebla, Mexico), and the following methodology was used to verify the sanitary quality of the lab-scale-manufactured cheeses. Counts of total mesophilic aerobic bacteria were completed using standard agar (Bioxon, BD, Edo. de Mexico, Mexico) following the method NOM-092-SSA1-1994 [43]; inoculated plates were incubated for 24 h at 37 ± 1.0 °C. For total coliform counts, violet-red bile agar (Bioxon, BD, Edo. de Mexico, Mexico) was used according to method NOM-111-SSA1-1994 [44], Baird–Parker agar (Bioxon, BD, Edo. de Mexico, Mexico) for *S. aureus* following the method NOM-115-SSA1-1994 [45], XLD agar (Bioxon, BD, Edo. de Mexico, Mexico) for *Salmonella,* Oxford agar (Difco, BD, Sparks, MD, USA) for *L. monocytogenes,* and MacConkey agar (Bioxon, BD, Edo. de Mexico, Mexico) for *E. coli*; plates were incubated at 37 ± 1.0 °C for 18–24 h. These determinations were performed in triplicate.

### 2.8. Statistical Analysis

The data of each experiment were assessed with analysis of variance (ANOVA) and Tukey’s mean comparison test *p* < 0.05. Minitab 20 software (Minitab LLC, State College, PA, USA) was utilized for the analysis.

## 3. Results and Discussion

### 3.1. Characterization of Lb. casei 21/1 CFS

Characterizing CFSs is essential in order to know the metabolites responsible for their antimicrobial activity. As shown in Table 1, the CFSs from MRS or WB had low pH values with significant differences (*p* < 0.05). The TA% values were high in comparison with conventional fermentations (1–2%) [46]. The values for the concentration of organic acids were high, with 1608.21 ± 11.62 and 866.42 ± 15.12 mM lactic acid for MRS and WB, respectively. In contrast, acetic acid values of 750.86 ± 17.80 mM for MRS and 146.83 ± 2.79 mM for WB were obtained. It must be considered that these values are high since the CFSs were concentrated, so the quantity of metabolites, such as organic acids, was increased. With this characterization, it is notable that the type of culture medium had a significant effect (*p* < 0.05), with the lab-culture medium (MRS) demonstrating a greater production of organic acids due to its excellent nutrient profile. In similar studies, lower concentrations (6.16 g L^−1^) of lactic acid than used in this study were reported in MRS fermentations with different strains of *Lacticaseibacillus casei* [47]. It is well known that acetic acid is a component that is produced in lesser quantities than lactic acid; acetic acid concentrations of 0.0128–0.0141 mmol mL^−1^ were reported in MRS broth fermented with *Lacticaseibacillus casei* BD 1415 [47].

Although the obtained values in the CFS from WB were significantly (*p* < 0.05) lower than those for MRS, *Lb. casei* produced enough organic acids in WB to exert antimicrobial activity. Whey fermentation media offers a sustainable, cost-effective, and environmentally friendly approach to producing lactic and acetic acid with LAB, making it a favorable choice for industrial applications. Whey fermentation by LAB is highly scalable and can be used for commercial production of lactic and acetic acid or as a fermentate for various applications as an antimicrobial agent [32,33]. For the food industry, whey is preferably used in powder form rather than liquid as this extends its shelf life and ease of handling. The whey powdering process consists of clarifying, separating cream, pasteurization, concentrating total solids by evaporation, lactose crystallization, and whey drying (commonly by spray drying) [30]. Most research related to whey focuses on its protein isolates and concentrates, which mainly contain β-lactoglobulin, α-lactalbumin, and bovine serum albumin, among others, and are used in industry for their high protein content and bioactive properties [48].

However, obtaining whey proteins substantially increases the price and involves high energy consumption. [48]. Therefore, all the components of whey have been used to design a sustainable, eco-friendly, and economically viable culture medium. This is an advantage compared with MRS broth, which has a high cost, and it has been reported that its use as a postbiotic or CFS in food products negatively affects organoleptic properties [19].

### 3.2. Physicochemical Analysis and Microbial Load of Cheese

The composition of the manufactured Mexican-style fresh cheese was moisture 63.42 ± 1.77%, protein 11.48 ± 1.49%, fat 5.40 ± 0.13%, ash 1.85 ± 0.03%, and 17.85% carbohydrates (by difference), while cheese yield was approximately 10%. Caro et al. [49] previously obtained different compositions for Panela cheese with lower moisture and lactose content (54.2% and 2.23%) and higher fat, protein, and ash content (18.8%, 18.4%, and 2.57%, respectively). The differences are attributed to the manufacturing process, in which the pressing force was probably insufficient to drain the whey.

Counts of the indicator microorganisms *E. coli*, *S.* enterica *serovar* Typhimurium, *S. aureus*, and *L. monocytogenes* on the uninoculated Mexican-style fresh cheese were under detection level (<10 CFU/g) (Table 2). The initial counts of total mesophilic aerobic bacteria were 2.54 ± 0.09 log_10_ CFU/g, and total coliforms were <10 CFU/g, indicating that these bacterial counts were acceptable for the aim of the study. However, microbiological analyses of commercially available Mexican-style fresh cheese samples obtained from local markets (Table 2) indicate that these cheeses presented results outside the established microbiological quality standards. These cheeses were not suitable for consumption and represent a risk to consumer health due to high bacterial counts and pathogenic microorganisms, including *L.* monocytogenes. The Mexican official standard NOM-243-SSA1-2010 [50] establishes a maximum allowed level for total coliform bacteria in milk derivatives of 2 log_10_ CFU g^−1^ or mL^−1^, aerobic mesophilic 5 log_10_ CFU g^−1^ or mL^−1^, and in the case of *S. aureus*, 2 log_10_ CFU g^−1^ or mL^−1^; the microbial counts of these commercially Mexican-style fresh cheeses were above the established maximum limits and may be associated with low hygiene in their food production and/or handling processes. Silva-Paz et al. [51] reported similar counts of aerobic mesophilic bacteria (≤5 to 6.5 ≥ log_10_ CFU g^−1^) and coliforms ranging from <100 to 5 ≥ log_10_ CFU g^−1^ in artisanal cheese.

Raw milk used in cheese production may contain pathogens such as *E. coli*, *Listeria*, *Salmonella*, and *S. aureus* [6,9,12]. These pathogens can survive and spread during cheese production if the milk is not pasteurized correctly or becomes contaminated during handling [52]. Cross-contamination can also occur if equipment, utensils, or surfaces are not properly sanitized. Proper hygiene practices during packaging, storage, and distribution of fresh cheese are essential to prevent pathogen contamination and ensure consumer safety.

### 3.3. Antimicrobial Effect of CFS in Mexican-Style Fresh Cheese

As can be observed in Figure 2, the storage time with the CFSs either from MRS or WB had a significant effect (*p* < 0.05) since a longer storage time (7 days) led to greater reductions in the microbial populations of indicator bacteria. The CFS from MRS broth decreased *S. enterica* serovar Typhimurium to 1.29 ± 0.05 log_10_ CFU g^−1^ after four days and 1.60 ± 0.04 log_10_ CFU g^−1^ after seven days of storage at 4 °C, while with CFS from WB, the microbial population decreased to 0.88 ± 0.08 log_10_ CFU g^−1^ after seven days. In the case of *L. monocytogenes*, applying the CFSs from MRS broth and WB medium decreased the counts by 1.43 ± 0.04 and 1.67 ± 0.03 log_10_ CFU g^−1^ after seven days of storage at 4 °C, respectively. Moreover, in vivo antimicrobial activity results showed that the type of culture medium affected the antimicrobial activity. The *S. enterica* serovar Typhimurium was the most sensitive microorganism to CFS from MRS broth, while *L. monocytogenes* was the most sensitive to the WB medium. *E. coli* and *S. aureus* were the most resistant to *Lb. casei* CFS (MRS and WB) activity in the studied Mexican-style fresh cheese. In a previous study, Arrioja-Bretón et al. [37] reported an in vitro assay for the antimicrobial activity of studied *Lb. casei* 21/1 CFS (from MRS) against the four tested indicator bacteria; *L. monocytogenes* was the most sensitive microorganism with inhibition zones higher than 22 mm, followed by *S. enterica* serovar Typhimurium, while the most resistant tested microorganisms were *S. aureus* and *E. coli* with lower inhibition halos.

The reductions in log_10_ cycles were similar to those reported in similar studies. For instance, Dal Bello et al. [21] successfully controlled the growth of *L. monocytogenes* using bacteriocins produced by *Lactococcus lactis* while manufacturing cottage cheese. In addition, Kousta et al. [53] studied the effect of adding ferulic acid or nisin (4 mg/g) to fresh cheese to inhibit *L. monocytogenes* growth; they observed reductions of 2.0 or 1.5 logarithmic cycles after 21 days of storage, respectively. Major inhibition rates were reported from *Lb. rhamnosus* in semi-hard goat cheese against *S. aureus* (21.66%), *L. monocytogenes* (10.23%), and *Salmonella enterica* serovar Enteritidis (5.52%) after 21 days of storage at 4 °C. [54]. Sometimes, LAB can only have a bacteriostatic effect, such as in the study reported by Settanni et al. [22], who controlled the growth of *L. monocytogenes* and *Salmonella* spp. in Tosèla cheese employing *Lactobacillus paracasei* NdP78 or *Streptococcus macedonicus* NdP1. They achieved reductions of 1.79 and 2.22 log_10_ CFU g^−1^ in coliforms and *S. aureus*, respectively.

The bacterial inhibition can be attributed to the high concentration of organic acids (lactic and acetic acid) in the CFSs, which act as antimicrobial compounds; their efficacy is well known since, in the food industry, organic acids are widely used as disinfectants or natural preservatives [16,46]. Organic acids reduce the pH of foods, and if this is <pKa of the organic acid, cell acidification occurs and interferes with the maintenance of the cell wall integrity of microorganisms, affecting the transport of metabolites and altering essential functions [46]. According to our previous reports, *Lb. casei*’s 21/1 antimicrobial activity is primarily due to organic acids, since neutralized CFS (pH 6.5) from MRS or flour–water mixture lost inhibitory activity entirely [37,55]. In the disinfection of the meat products, lactic acid has been widely used, with concentrations close to that of this study (1342.15 mM or ≈10% *v*/*w*) applied at 55 °C on bovine skin reducing *Salmonella* population to 3.4 log_10_ cycles [56]. Meanwhile, acetic acid was used to disinfect meat at concentrations of 268.43 mM or ≈2% *v*/*w*, reducing the microbial population of *S. enterica* serovar Typhimurium and *E. coli* [46]. As observed in Figure 3, the pH of the cheeses decreased significantly (*p* < 0.05) with the addition of CFSs (4.23–4.60). The pH of the control cheese was in the range reported for queso blanco or queso fresco (5.2–6.8) [57]. Previous studies also observed a descending pH value of foods when CFS from *Lactobacillus* was applied. For instance, fresh beef surfaces treated with CFS from *Lb. plantarum* reached a pH of 3.80 [37]. In contrast, ground beef’s pH decreased by 0.5 or 1.0 after 3 or 6 days of storage at 4 °C (pH = 4.9) wrapped in bacterial nanocellulose films impregnated with 20% CFS from *Lb. plantarum* [58]. Lactic and acetic acids are impregnated and incorporated into the food matrix, leading to a lower pH after CFS treatment. Regarding storage, the pH remained stable and no significant differences were observed (*p* > 0.05). With these results, it can be argued that CFS from WB medium fermented with *Lb. casei* 21/1 is a source of antimicrobial compounds that are sustainable, eco-friendly, and economically viable. Furthermore, applying them as bio-preservatives to Mexican-style fresh cheeses would help avoid foodborne diseases and improve food safety, given that it has been demonstrated that commercial cheeses present a risk for human consumption.

### 3.4. Appearance of Cheese with CFSs of Lb. casei 21/1

The appearance and color of a product are important aspects to consider since they influence consumers’ acceptability. As can be seen (Figure 4), the addition of CFS from MRS affected the appearance of the cheese; a slight brown color developed, while CFS from WB media did not exert a notable change. This agrees with the color parameters (Table 3), because the differences in color to MRS samples were more distinctive than in the WB-treated cheeses, according to the scale categorization proposed by Francis and Clydesdale [59]. Furthermore, the contribution of a* in MRS cheeses increased significantly (*p* < 0.05), and L* values decreased significantly (*p* < 0.05), giving brownish colors characteristic of the culture medium. On the other hand, the parameters of color in cheeses treated with CFS from WB presented an increase in yellowish color due to the a* and b* values, which significantly (*p* < 0.05) increased and decreased, respectively. The brown color of CFS from MRS broth directly impacted the food color, whereas WB medium slightly modified it. In previous studies, CFS from MRS stained beef cubes (wrapped in whey isolate film with added CFS from *L. sakei* or immersed in a marinade containing CFS from *L. plantarum*) to brown colors [55,60].

## 4. Conclusions

The studied cell-free supernatants from *Lb. casei* 21/1 from MRS broth and WB medium were effective against microorganisms causing foodborne illness, mainly *S. enterica* serovar Typhimurium and *L. monocytogenes*. The CFS from WB medium can be a suitable alternative as a bio-preservative in foods at risk of contamination, such as Mexican-style fresh cheeses, due to its economic viability and as a safe consumption ingredient. The incorporation of CFS from WB medium slightly affected the color and appearance of cheeses compared with MRS broth. Further studies could examine the best way to incorporate supernatants into cheeses or other foods, evaluating the contact time and changes in sensory properties. Although *Lb. casei* 21/1 is a LAB commonly recognized as GRAS, it will be important to consider toxicological studies of CFS obtained from the WB culture medium. In addition, exploration of other bioactive properties that benefit consumers’ health could be sought, as occurs with postbiotics. Whey, a by-product of cheese production, can be utilized as a fermentation substrate, which can help reduce production costs compared with conventional culture medium. Not only is this cost effective, but it also aids in managing dairy waste sustainably. Whey contains valuable nutrients that promote the growth of lactic acid bacteria, thus enhancing acid production efficiency. Whey fermentation aligns with sustainability and regulatory guidelines by reducing waste and promoting efficient resource use, helping companies comply with environmental regulations and meet their social responsibility objectives.

## Figures and Tables

**Figure 1 ijerph-21-00560-f001:**
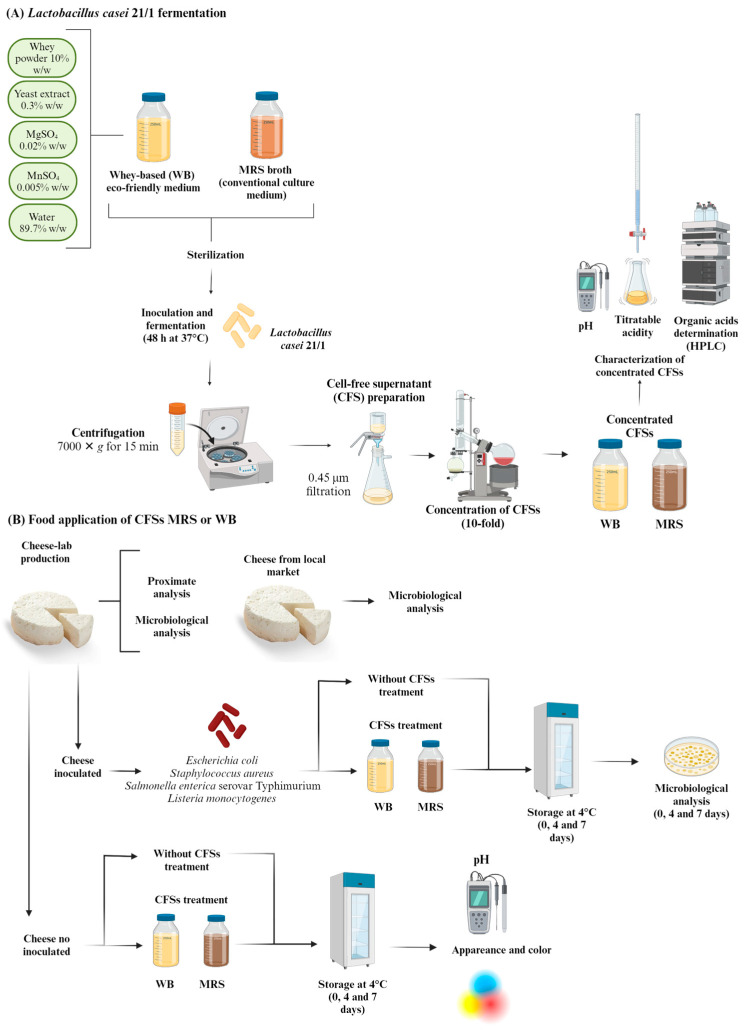
Flow diagram of the experiments, materials, and methods used for the present study.

**Figure 2 ijerph-21-00560-f002:**
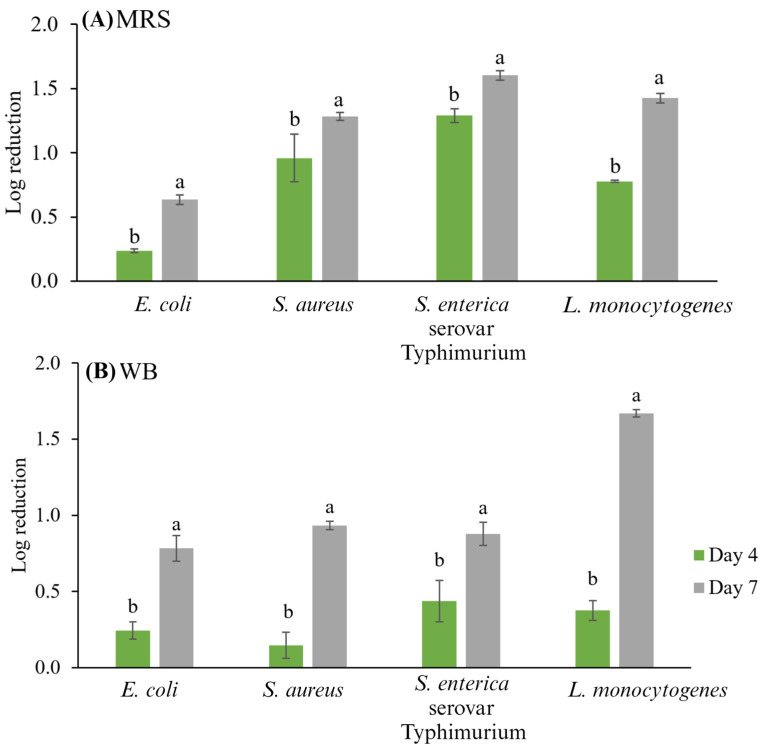
Logarithmic reductions (N/N_0_) of tested indicator microorganisms (*E. coli* ATCC 25922, *S. aureus* ATCC 29213, *S. enterica* serovar Typhimurium ATCC 14028, and *L. monocytogenes* Scott A) in studied Mexican-style fresh cheese treated with *Lb. casei* 21/1 CFS from (**A**) de Man, Rogosa, and Sharpe (MRS) broth and (**B**) whey-based medium (WB). Different letters show a significant difference (*p* < 0.05) between the samples.

**Figure 3 ijerph-21-00560-f003:**
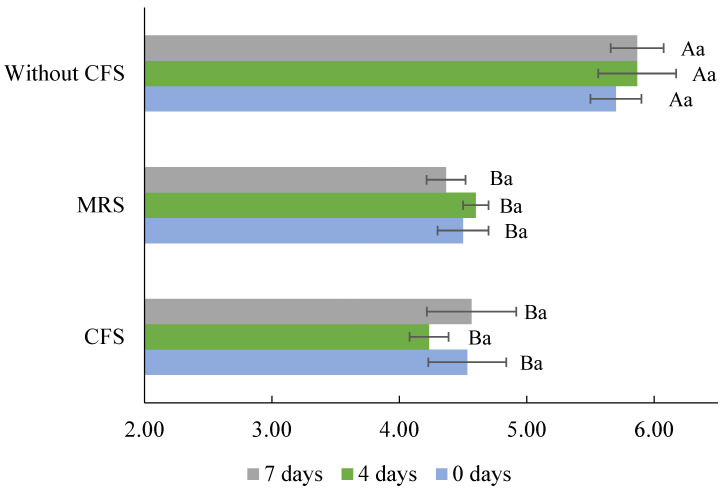
Evolution of pH during storage of fresh Mexican-style cheeses without cell-free supernatant (Without CFS) and with *Lb. casei* 21/1 CFS from de Man, Rogosa, and Sharpe broth (MRS) or whey medium (CFS). Different capital letters indicate significant differences between the different cheeses at the same storage time at 4 °C (*p* < 0.05). Different lowercase letters indicate significant differences (*p* < 0.05) between storage time at 4 °C in the same sample.

**Figure 4 ijerph-21-00560-f004:**
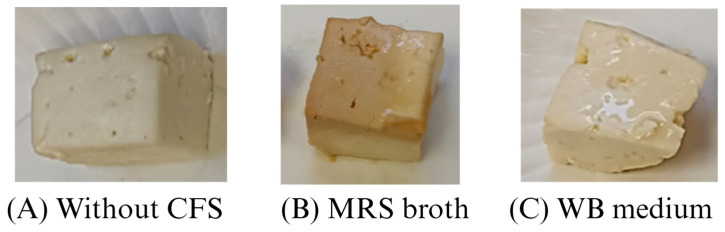
Comparison of cheeses after 7 days of storage at 4 °C treated (**A**) without cell-free supernatant (CFS), (**B**) MRS broth, and (**C**) WB medium obtained from *Lb. casei* 21/1.

**Table 1 ijerph-21-00560-t001:** pH, titratable acidity, and acids of concentrated *Lb. casei* 21/1 cell-free supernatants (CFSs) from de Man, Rogosa, and Sharpe (MRS) broth and whey-based (WB) medium.

	MRS	WB
pH	3.87 ± 0.03 ^a^	3.47 ± 0.06 ^b^
TA%	11.44 ± 0.50 ^a^	6.14 ± 0.29 ^b^
Lactic acid (mM)	1608.21 ± 11.62 ^a^	866.42 ± 15.12 ^b^
Acetic acid (mM)	750.86 ± 17.80 ^a^	146.83 ± 2.79 ^b^

TA%: Percentage of titratable acidity (expressed as lactic acid). Different letters in the same row show a significant difference (*p* < 0.05) between CFSs.

**Table 2 ijerph-21-00560-t002:** Microbial counts (log_10_ CFU/g) of Mexican-style fresh cheeses.

Cheese	*E. coli*	*L. monocytogenes*	*S. aureus*	Aerobic Mesophilic Bacteria	Total Coliforms
Local market 1	2.05 ± 0.16 ^a^	1.11 ± 0.10 ^a^	1.64 ± 0.03 ^b^	7.85 ± 0.05 ^a^	2.23 ± 0.46 ^a^
Local market 2	2.13 ± 0.10 ^a^	1.27 ± 0.10 ^a^	1.80 ± 0.01 ^a^	4.70 ± 0.01 ^b^	2.18 ± 0.04 ^b^
Manufactured in laboratory	<10 ^c^	<10 ^c^	<10 ^c^	2.54 ± 0.09 ^c^	<10 ^c^

Different letters show a significant difference (*p* < 0.05) between the analyzed cheeses.

**Table 3 ijerph-21-00560-t003:** Color properties and color difference of Mexican-style fresh cheeses with *Lb. casei* 21/1 CFS.

Time	Control (without CFS)			
	*L**	*a**	*b**	ΔE
0	89.98 ± 0.8 ^Ba^	2.65 ± 0.01 ^Ab^	13.53 ± 0.23 ^Bb^	
4	90.30 ± 0.23 ^Ba^	2.85 ± 0.03 ^Aa^	14.18 ± 0.04 ^Ca^	0.26 ± 0.05
7	90.16 ± 0.03 ^Aa^	2.79 ± 0.09 ^Aa^	13.45 ± 0.08 ^Cb^	0.77 ± 0.10
	MRS cell-free supernatant			
0	82.78 ± 0.08 ^Ca^	−0.05 ± 0.13 ^Cb^	20.93 ± 0.05 ^Aa^	10.67 ± 0.03
4	77.51 ± 0.09 ^Cb^	2.11 ± 0.03 ^Ba^	18.12 ± 0.13 ^Bb^	13.29 ± 0.04
7	70.74 ± 0.3 ^Bc^	2.14 ± 0.04 ^Ba^	19.29 ± 0.04 ^Ac^	20.09 ± 0.27
	Whey-based cell-free supernatant			
0	93.75 ± 0.14 ^Aa^	1.6 ± 0.27 ^Ba^	9.1 ± 0.22 ^Cc^	5.92 ± 0.19
4	91.91 ± 0.31 ^Ab^	−2.94 ± 0.04 ^Cb^	19.71 ± 0.67 ^Aa^	5.36 ± 0.04
7	90.46 ± 0.21 ^Ac^	−1.20 ± 0.05 ^Cc^	17.23 ± 0.1 ^Bb^	8.57 ± 0.48

Color CIELAB scale parameters: *L**: luminosity, *a**: red–green contribution, *b**: blue–yellow contribution, ΔE: color differences. Different capital letters indicate significant differences between the different cheeses at the same storage time at 4 °C (*p* < 0.05). Different lowercase letters indicate significant differences (*p* < 0.05) between storage time at 4 °C in the same sample.

## Data Availability

The data presented in this study are available on request.

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
