# Peer review of "Utilization of Whey for Eco-Friendly Bio-Preservation of Mexican-Style Fresh Cheeses: Antimicrobial Activity of Lactobacillus casei 21/1 Cell-Free Supernatants (CFS)"

_ijerph, 2024, doi:10.3390/ijerph21050560_

Round 1
Reviewer 1 Report
Comments and Suggestions for Authors
This article is well written on (Utilization of Whey for Eco-Friendly Bio-Preservation of Mexi- 2 can-style fresh cheeses: Antimicrobial Activity of Lactobacillus 3 casei 21/1 Cell-Free Supernatants (CFS)), Nonetheless, if the writers implement these recommendations, the manuscript's quality will improve.
1. The abstract needs some improvements that should summarize the entire text better. Authors should provide the importance of the study, and Short description about all analysis or at least main points of methodology.
2. The authors did not mention any statistical analysis in the abstract section please provide the statistical design and analysis that you have used.
3. Keywords should be in alphabetical order.
4. In introduction should give the basic back ground related to utilization of Dairy industry by-products and then they should must provide some latest studies about the whey obtained from dairy industry as by-product as some recent studies have been conducted on this emerging topic that should must be discuss here.
5. In section 2.3.2. Organic acid determination in CFS please mention which product was used to analyze the organic acids.
6. Please provide references in all the methods like in line 127 and section 2.4. cheese production which method was adopted. Similarly no references in physicochemical analysis except proximate analysis.
7. Authors should provide a paragraph that should comprise of rationale and aims of the study in the end of introduction section.
8. Authors should provide a flow diagram of whole process including whey protein, and its purification procedure.
9. Authors should also give the flow diagram in methods that should explain the complete process of these activities and also mention how these are eco-friendly as well for the easy understanding of the readers.
10. In the manuscript, list the full form first, then abbreviations.
11. Keep the beginning with recent supportive findings, and if you can find a 2022, 2023, 2024 main title introduction, that would be preferable.
12. Find a more recent reference than “Moreno-Enriquez, R. I.; Garcia-Galaz, A.; Acedo-Felix, E.; Gonzalez-Rios, H.; Call, J. E.; Luchansky, J. B.; Diaz-Cinco, M. E. 353 Prevalence, Types, and Geographical Distribution of Listeria Monocytogenes from a Survey of Retail Queso Fresco and Associated 354 Cheese Processing Plants and Dairy Farms in Sonora, Mexico†. J. Food Prot. 2007, 70 (11), 2596–2601. https://doi.org/10.4315/0362- 355 028X-70.11.2596.”
13. “Villar, R. G. Investigation of Multidrug-Resistant Salmonella Serotype Typhimurium DT104 Infections Linked to Raw-Milk 362 Cheese in Washington State. JAMA 1999, 281 (19), 1811. https://doi.org/10.1001/jama.281.19.1811.” Try to find recent.
14. Find a more recent reference than “Pastore, R.; Schmid, H.; Altpeter, E.; Baumgartner, A.; Hächler, H.; Imhof, R.; Sudre, P.; Boubaker, K. Outbreak of Salmonella 364 Serovar Stanley Infections in Switzerland Linked to Locally Produced Soft Cheese, September 2006 – February 2007. Eurosur- 365 veillance 2008, 13 (37). https://doi.org/10.2807/ese.13.37.18979-en”
15. Most references are old, reducing their visual appeal. Try to provide recent data.
16. 12. The manuscript has a few superscript references like [23] and a few conventional ones [23]. Please recheck it according to journal requirement.
17. Figure 3 (a,b,c) are quite blur. The blurriness of Figure 3 is evident. Attempt to incorporate a comprehensible visual representation.
18. Improve grammatical mistakes in the whole manuscript.
19. Authors have not adequately justified results and compared them to previous studies in the discussion section.
20. Authors should properly justify and give comparison with other studies and mention that how their findings are significant than others.
21. Consider ways to improve conclusion section. Do your findings offer physicians recommendations? Can you recommend a clinically effective formulation or examine it in future studies? Can you advise other researchers in your field?
22. Improve references according to journal requirements.
Comments on the Quality of English LanguageExtensive editing of English language required
Author Response
Review Report (Reviewer 1)
This article is well written on (Utilization of Whey for Eco-Friendly Bio-Preservation of Mexi- 2 can-style fresh cheeses: Antimicrobial Activity of Lactobacillus 3 casei 21/1 Cell-Free Supernatants (CFS)), Nonetheless, if the writers implement these recommendations, the manuscript's quality will improve.
Thank you for your time reviewing our manuscript and for your comments and suggestions.
- The abstract needs some improvements that should summarize the entire text better. Authors should provide the importance of the study, and Short description about all analysis or at least main points of methodology.
R=The abstract was modified to summarize the methodology and add the study's main findings.
- The authors did not mention any statistical analysis in the abstract section please provide the statistical design and analysis that you have used.
R= We added in the abstract the significant differences (p < 0.05) found in the analyzed parameters during the study.
- Keywords should be in alphabetical order.
R= The keywords were ordered in alphabetical order.
- In introduction should give the basic back ground related to utilization of Dairy industry by-products and then they should must provide some latest studies about the whey obtained from dairy industry as by-product as some recent studies have been conducted on this emerging topic that should must be discuss here.
R= We added more background about the whey used in the food industry, whey as a medium culture for LAB, and previous studies about the antimicrobial properties of fermented whey by LAB.
- In section 2.3.2. Organic acid determination in CFS please mention which product was used to analyze the organic acids.
R= We mentioned the samples analyzed for organic acids.
- Please provide references in all the methods like in line 127 and section 2.4. cheese production which method was adopted. Similarly, no references in physicochemical analysis except proximate analysis.
R= References are added for the cheese-making process and physicochemical analyses carried out on the cheeses.
- Authors should provide a paragraph that should comprise of rationale and aims of the study in the end of introduction section.
R= The final paragraph of the introduction was modified and separated to state the objective of our research appropriately.
- Authors should provide a flow diagram of the whole process including whey protein, and its purification procedure.
R= We did not utilize whey protein (isolate or concentrate) for the present study, but we clarified the use of whey powder (See 3.1 section).
- Authors should also give the flow diagram in methods that should explain the complete process of these activities and also mention how these are eco-friendly as well for the easy understanding of the readers.
R=We included a flow diagram (Figure 1) explaining the experiments performed and the materials and methods used in the present investigation. In addition, we improved the information about the benefits of using whey as a culture medium in section 3.1.
- In the manuscript, list the full form first, then abbreviations.
R= We revised and corrected the full name of the abbreviations of the manuscript.
- Keep the beginning with recent supportive findings, and if you can find a 2022, 2023, 2024 main title introduction, that would be preferable.
R= More recent studies about the topic were replaced (deleting the old ones) and added.
- Find a more recent reference than “Moreno-Enriquez, R. I.; Garcia-Galaz, A.; Acedo-Felix, E.; Gonzalez-Rios, H.; Call, J. E.; Luchansky, J. B.; Diaz-Cinco, M. E. 353 Prevalence, Types, and Geographical Distribution of Listeria Monocytogenes from a Survey of Retail Queso Fresco and Associated 354 Cheese Processing Plants and Dairy Farms in Sonora, Mexico†. J. Food Prot. 2007, 70 (11), 2596–2601. https://doi.org/10.4315/0362- 355 028X-70.11.2596.”
R= We change the reference to a more recent one.
- “Villar, R. G. Investigation of Multidrug-Resistant Salmonella Serotype Typhimurium DT104 Infections Linked to Raw-Milk 362 Cheese in Washington State. JAMA 1999, 281 (19), 1811. https://doi.org/10.1001/jama.281.19.1811.” Try to find recent.
R= We change the reference to a more recent one.
- Find a more recent reference than “Pastore, R.; Schmid, H.; Altpeter, E.; Baumgartner, A.; Hächler, H.; Imhof, R.; Sudre, P.; Boubaker, K. Outbreak of Salmoella 364 Serovar Stanley Infections in Switzerland Linked to Locally Produced Soft Cheese, September 2006 – February 2007. Eurosur- 365 veillance 2008, 13 (37). https://doi.org/10.2807/ese.13.37.18979-en”
R= We change the reference to a more recent one.
- Most references are old, reducing their visual appeal. Try to provide recent data.
A= We review current publications and change old references for more current ones.
- 12. The manuscript has a few superscript references like [23] and a few conventional ones [23]. Please recheck it according to journal requirement.
R= The format to report the references was revised and corrected throughout the entire manuscript.
- Figure 3 (a,b,c) are quite blur. The blurriness of Figure 3 is evident. Attempt to incorporate a comprehensible visual representation.
R= We added other photos to replace the blurry ones in Figure 3.
- Improve grammatical mistakes in the whole manuscript.
R= The entire manuscript was reviewed, grammatical errors were corrected, and in some paragraphs (e.g., section 3.3), the text and its coherence were modified to facilitate the reading of the manuscript.
- Authors have not adequately justified and compared results to previous studies in the discussion section.
R = We review the results and their discussion and include appropriate comparisons with other similar studies.
- Authors should properly justify and give comparison with other studies and mention that how their findings are significant than others.
R= Similar to your previous comment, the study's main findings were highlighted and compared with others relevant to this topic.
- Consider ways to improve conclusion section. Do your findings offer physicians recommendations? Can you recommend a clinically effective formulation or examine it in future studies? Can you advise other researchers in your field?
R= We made some changes in the conclusions, commented on the importance of clinical and toxicological studies, and discovered other bioactive properties of CFSs.
- Improve references according to journal requirements.
R= R= The format of citations and references was revised and corrected throughout the entire manuscript.
Reviewer 2 Report
Comments and Suggestions for Authors
The paper entitled “Utilization of Whey for Eco-Friendly Bio-Preservation of Mexican-style fresh cheeses: Antimicrobial Activity of Lactobacillus casei 21/1 Cell-Free Supernatants (CFS)” focuses on the exploring the use of whey as a sustainable resource to reduce waste and environmental impact in the food industry. Authors investigated the antimicrobial effects of CFS from Lactobacillus casei 21/1 produced in two different media: MRS and WB. The manuscript is in the aims and scope of IJERPH journal. The manuscript is well written and clear. The experiments are written in a good and reproducible way. However, it needs to be improved to match the quality standards of the IJERPH journal.
L34-L36: Put the reference at the end of the phrase.
L41-L43: “It is not uncommon..production”. Add references.
L50-L53: “LABs…feed” put the references at the end of the phrase. This will help the reader. Do the same in all the manuscripts.
L60: “Generally regarded as safe” is GRAS from the FDA? Because in the FDA, the acronym refers to “Generally Recognized as Safe”.
The layout of the Table 2 must be improved.
Author Response
Review Report (Reviewer 2)
The paper entitled “Utilization of Whey for Eco-Friendly Bio-Preservation of Mexican-style fresh cheeses: Antimicrobial Activity of Lactobacillus casei 21/1 Cell-Free Supernatants (CFS)” focuses on the exploring the use of whey as a sustainable resource to reduce waste and environmental impact in the food industry. Authors investigated the antimicrobial effects of CFS from Lactobacillus casei 21/1 produced in two different media: MRS and WB. The manuscript is in the aims and scope of IJERPH journal. The manuscript is well written and clear. The experiments are written in a good and reproducible way. However, it needs to be improved to match the quality standards of the IJERPH journal.
Thank you for your time reviewing our manuscript and for your comments and suggestions.
L34-L36: Put the reference at the end of the phrase.
R= We put the reference at the end of the phrase.
L41-L43: “It is not uncommon..production”. Add references.
R= A reference was added.
L50-L53: “LABs…feed” put the references at the end of the phrase. This will help the reader. Do the same in all the manuscripts.
R= We revised the entire manuscript and put the references at the end of the sentences.
L60: “Generally regarded as safe” is GRAS from the FDA? Because in the FDA, the acronym refers to “Generally Recognized as Safe”.
R= We change the word regarded by recognized.
The layout of the Table 2 must be improved.
R= We modified Table 2 in a different orientation to improve the layout.
Reviewer 3 Report
Comments and Suggestions for Authors
The manuscript submitted by Vera-Santander et al. aimed to investigate the application of supernatant from L. casei cultivated in two distinct mediums (one of them being whey, a byproduct that requires better utilization in the dairy industry) in experimentally contaminated cheeses during storage. The research is highly interesting and holds significant relevance for the dairy sector, as it employs a production residue for the generation of components that extend the shelf life of dairy products.
The introduction is impeccably written, providing a comprehensive contextualization of the study. However, the Materials and Methods section requires adjustments to enhance comprehension (as outlined in the remarks). Additionally, I suggest the inclusion of a flowchart depicting the experimental design in a clear and organized manner to accurately depict the procedures conducted by the authors. This would significantly improve the understanding of the research.
The Results and Conclusion sections are well-presented and appropriate. Nevertheless, the Results section proves challenging to comprehend due to the absence of a clear delineation of the experimental design, as mentioned earlier.
Remarks:
Avoid using keywords already present in the title. Replace them with terms that provide better contextualization of the study area for improved indexing.
Line 53: Reference studies on different foods.
Line 156: Check the abbreviation of the genera for all scientific names throughout the manuscript and verify the taxonomic norms accordingly (e.g., S. aureus, Lb casei).
In section 2.6, include descriptions of the inoculants: how were they prepared and quantified? The authors only mention their use and the expected CFU per surface but fail to describe the methodologies employed in preparation and quantification. Was a non-contaminated control cheese included?
Line 166: Concentration of peptone water.
Lines 176-177: Clarify the reasons behind this. It needs to be explicit. In fact, I fail to comprehend why this was done. I suggest removing this part from the Materials and Methods section and also from the Results. It does not contribute to the study.
The Methodology section requires more details regarding the experimental design, which is quite incomprehensible. While the authors delve into the methodologies used, the design itself is not clear enough. For instance, I couldn't discern how many cheeses were produced and contaminated with each microorganism, and the number of study repetitions.
Comments on the Quality of English LanguageMinor editing.
Author Response
Review Report (Reviewer 3)
The manuscript submitted by Vera-Santander et al. aimed to investigate the application of supernatant from L. casei cultivated in two distinct mediums (one of them being whey, a byproduct that requires better utilization in the dairy industry) in experimentally contaminated cheeses during storage. The research is highly interesting and holds significant relevance for the dairy sector, as it employs a production residue for the generation of components that extend the shelf life of dairy products.
The introduction is impeccably written, providing a comprehensive contextualization of the study. However, the Materials and Methods section requires adjustments to enhance comprehension (as outlined in the remarks). Additionally, I suggest the inclusion of a flowchart depicting the experimental design in a clear and organized manner to accurately depict the procedures conducted by the authors. This would significantly improve the understanding of the research.
The Results and Conclusion sections are well-presented and appropriate. Nevertheless, the Results section proves challenging to comprehend due to the absence of a clear delineation of the experimental design, as mentioned earlier.
Thank you for your time reviewing our manuscript and for your comments and suggestions.
Remarks:
Avoid using keywords already present in the title. Replace them with terms that provide better contextualization of the study area for improved indexing.
R= The keywords were modified and replaced by Cell-free supernatants, dairy industry, food safety, natural antimicrobials, lactic acid bacteria, and sustainability.
Line 53: Reference studies on different foods.
R= We include references that focus on the biopreservation of food by LAB.
Line 156: Check the abbreviation of the genera for all scientific names throughout the manuscript and verify the taxonomic norms accordingly (e.g., S. aureus, Lb casei).
R= We revised some taxonomic norms and changed S. Typhimurium to S. enterica serovar Typhimurium, leaving Staphylococcus aureus as S. aureus, Lactobacillus casei 21/1 as Lb. casei 21/1, and Listeria monocytogenes as L. monocytogenes.
In section 2.6, include descriptions of the inoculants: how were they prepared and quantified? The authors only mention their use and the expected CFU per surface but fail to describe the methodologies employed in preparation and quantification. Was a non-contaminated control cheese included?
R= We include more details about preparing inoculums, either LAB or indicator bacteria, and the methodology employed to inoculate the indicator bacteria in the cheese.
Line 166: Concentration of peptone water.
R= The concentration of peptone water was added.
Lines 176-177: Clarify the reasons behind this. It needs to be explicit. In fact, I fail to comprehend why this was done. I suggest removing this part from the Materials and Methods section and also from the Results. It does not contribute to the study.
R= The reasons for carrying out these analyses are explained more clearly, as well as the purpose of the comparison between cheeses from local markets and those made in the laboratory.
The Methodology section requires more details regarding the experimental design, which is quite incomprehensible. While the authors delve into the methodologies used, the design itself is not clear enough. For instance, I couldn't discern how many cheeses were produced and contaminated with each microorganism, and the number of study repetitions.
R= To clarify the experiments performed, we added a flow diagram or scheme in materials and methods. Also, include the details about repetitions in each test.
Round 2
Reviewer 1 Report
Comments and Suggestions for Authors
Authors have improved well, but still there are some grammatical errors that should be rechecked.
Comments on the Quality of English LanguageEnglish check is required again.
Reviewer 3 Report
Comments and Suggestions for Authors
This mamuscript was improved.